

# Population structure and microbial community diversity of two common tetillid sponges in a tropical reef lagoon

Jake Ivan P. Baquiran, Michael Angelou L. Nada, Niño Posadas, Dana P. Manogan, Patrick C. Cabaitan and Cecilia Conaco

Marine Science Institute, University of the Philippines Diliman, Quezon City, Philippines

## ABSTRACT

Sponges are predicted to dominate future reef ecosystems influenced by anthropogenic stressors and global climate change. The ecological success of sponges is attributed to their complex physiology, which is in part due to the diversity of their associated prokaryotic microbiome. However, the lack of information on the microbial community of many sponge species makes it difficult to gauge their interactions and functional contributions to the ecosystem. Here, we investigated the population dynamics and microbial community composition of two tetillid sponges identified as *Cinachyrella* sp. and *Paratetilla* sp., which are common on coral bommies in a reef lagoon in Bolinao, northwestern Philippines. The sponges ranged in size from $2.75 \pm 2.11$ to $6.33 \pm 3.98$ cm (mean $\pm$ standard deviation) and were found at an average density of $1.57 \pm 0.79$ to $4.46 \pm 3.60$ individuals per sq. m. on the bommies. The tetillid sponge population structure remained stable over the course of four years of monitoring. Prokaryotic communities associated with the sponges were distinct but had overlapping functions based on PICRUSt2 predictions. This convergence of functions may reflect enrichment of metabolic processes that are crucial for the survival of the tetillid sponges under prevailing conditions in the reef lagoon. Differentially enriched functions related to carbon, sulfur, fatty acid, and amino acid metabolism, cellular defense, and stress response, may influence the interactions of tetillid sponges with other biota on the bommies.

## INTRODUCTION

Sponges (phylum Porifera) are a diverse group of sessile, filter-feeding invertebrate animals. They are a major component of benthic ecosystems and are responsible for many ecological processes, such as ecosystem structuring via reef consolidation and bio-erosion (*Bell, 2008*). Sponges link the whole reef system through the 'sponge loop' whereby dissolved organic matter released by benthic primary producers is made available to higher trophic levels in the form of particulate detritus (*De Goeij et al., 2013*). Sponges are also consumed as food by some spongivores (e.g., parrotfishes, angelfishes) (*Wooster, Marty & Pawlik, 2016*) and they offer refuge for juvenile commensal invertebrates (*Ribeiro, Omena & Muricy, 2003*), reef

Corresponding author
Cecilia Conaco,
cconaco@msi.upd.edu.ph

fish recruits (*Cabaitan, Gomez & Yap, 2016*), and other macroflora (*Di Camillo et al., 2017*).

Sponges are predicted to be winners under future ocean conditions brought about by the changing climate (*Bell et al., 2013*). Although studies to date are limited to a few species, some sponges have been shown to have lower sensitivity to elevated seawater temperature and ocean acidification, while others appear to benefit from the combined effects of these stressors (*Bell et al., 2018*). For example, sponge populations in Brazil remained stable even under elevated seawater temperatures brought by the El Niño Southern Oscillation event (*Kelmo, Bell & Attrill, 2013*). Exposure of other sponge species to combined ocean warming and acidification showed no effect on growth, survival, or secondary metabolite production (*Duckworth et al., 2012*). In fact, the bioeroding sponge, *Cliona orientalis*, even exhibited increased biomass and faster bioerosion rates under these conditions (*Fang et al., 2013*). The tolerance of sponges to environmental adversities might be attributed to the host, as well as to their associated prokaryotic symbionts.

Sponges are holobionts that are associated with a diverse array of microorganisms (*Thomas et al., 2016*). These symbionts are essential for nutrition, immunity, defense, and reproduction of the sponge host (*Reiswig, 1975*; *Pita et al., 2018*). Sponge-associated prokaryotes are predicted to have the capability for a wide range of metabolic processes, including photosynthesis, nitrogen fixation, ammonium oxidation, sulfate reduction, and sulfur oxidation (*Hoffman et al., 2005*; *Pita et al., 2018*; *Feng & Li, 2019*). The diverse microbial community in sponges may contribute to the ecological adaptability and plasticity of the holobiont, allowing it to thrive even in perturbed environments (*Erpenbeck et al., 2016*; *Bang et al., 2018*). However, the stability of sponge microbial communities can vary among host species and under different environmental conditions. For example, the microbiomes of *Cymbastela stipitata* and *Gelliodes obtusa* remained stable even under eutrophication stress (*Luter, Gibb & Webster, 2014*; *Baquiran & Conaco, 2018*). Similarly, the microbiome of the Great Barrier Reef sponge, *Rhopaloeides odorabile,* did not change under eutrophication and elevated temperature conditions (*Simister et al., 2012*). On the other hand, raising seawater temperature past its tolerance threshold disrupted the microbiome of *C. orientalis* (*Ramsby et al., 2018*). Light attenuation treatments mimicking the effect of dredging activities also caused a shift in the bacterial community of phototrophic sponges but did not affect heterotrophic sponges (*Pineda et al., 2016*).

While next-generation sequencing approaches have begun to uncover the diversity of sponge associated prokaryotes, the lack of baseline data on the microbial community composition of most sponge species makes it difficult to assess the interactions between microbes and their hosts, as well as the functional contributions of marine sponges at larger ecological scales. This emphasizes the need to better understand the diversity of sponges and sponge-associated microbes and to identify microbially-driven functions in sponges to gain a more comprehensive understanding of the processes within the sponge holobiont that bear implications on ecosystem functions and biogeochemical cycles.

Tetillid sponges are classified under family Tetillidae of order Tetractinellida, class Demospongiae. They generally possess a globular morphology with crater-like depressions (*Rützler & Smith, 1992*) and are commonly referred to as "moon sponges" (*Chambers et al., 2013*; *Santodomingo & Becking, 2018*). The pronounced circular configuration of their

megascleres, minimal basal attachment, and almost solid spicule core allow tetillid sponges to inhabit environments that are influenced by frequent disturbances (*Byrne, 1987*). Tetillids serve as important structural constituents of reef systems where they provide habitats and other functions for many organisms (*McDonald, Hooper & McGuinness, 2002*; *Van Soest & Rutzler, 2002*). Tetillid sponges are a challenge to identify visually in the field, particularly for individuals from closely related taxa or from cryptic sympatric populations (*Szitenberg et al., 2013*). However, studies have shown that these sponges may be differentiated based on their distinct microbial community compositions (*Chambers et al., 2013*; *Cuvelier et al., 2014*).

This present work aims to elucidate the population density, size frequency distribution, and prokaryotic microbial community composition of common tetillid sponges on coral bommies in a tropical reef lagoon in Bolinao, northwestern Philippines. This site is influenced by multiple stressors, including rising sea surface temperatures, increased precipitation, and frequent typhoons (*Dado & Takahashi, 2017*; *Fang et al., 2006*; *Peñaflor et al., 2009*). Nutrient loading due to submarine groundwater discharge and nutrient plumes extending from a nearby mariculture zone is also a persistent condition (*San Diego-McGlone et al., 2008*; *Senal et al., 2011*; *Udarbe-Walker & Magdaong, 2003*). The combined effect of these stressors has resulted in several bleaching events that has led to reduced live coral cover (*Cabaitan, Gomez & Yap, 2016*; *Gurney et al., 2013*), yet sponges like the tetillids are prevalent in the area.

## MATERIAL AND METHODS

### Study site

The study was conducted on five coral bommies (Fig. S1) within the lagoon of the Santiago reef flat in Bolinao, northwestern Philippines (B15: 16°25′50.7″N, 119°55′02.1″E; B16: 16°25′50.6″N, 119°55′07.9″E; B19: 16°25′47.8″N, 119°55′14.1″E; B21: 16°25′48.6″N, 119°55′21.0″E; B22: 16°25′50.2″N, 119°55′24.5″E). The bommies range from 20 to 60 m in diameter and are distributed across a distance of about 500 m. The bommies are located about 200 m north from a populated area on Santiago Island and about 400 m south of the unpopulated side of Silaqui Island. To the west of the bommies is the South China Sea or West Philippine Sea while to the east is the Lingayen Gulf. The bommies are 7–10 km from the mariculture zone in the Guiguiwanen channel to the south of Santiago Island. The organic matter and nutrient-enriched plume from this zone can be driven by currents around Santiago Island towards the lagoon where the bommies are located (*Udarbe-Walker & Magdaong, 2003*). In addition, submarine groundwater discharge may be a significant source of nutrients in the reef flat (*Senal et al., 2011*).

### Field surveys

A transect was laid at the base around each bommie about a meter above the sandy substrate. A 1 sq. m. quadrat was placed every 2 m along each transect. All tetillid individuals found inside each quadrat were counted and photographed. Sponge sizes were determined from the images using Coral Point Count with Excel extensions or CPCe (*Kohler & Gill, 2006*). Field surveys were conducted in May 2016, August 2017, September 2018, and July 2019.

A separate field survey was conducted in September 2019 where small tissue cores were taken from all tetillid sponges within each quadrat to estimate the abundance of species based on their characteristic internal tissue color.

## Measurement of environmental parameters

Environmental parameters were collected at set points around the bommies during each field survey event. A multi-parameter meter (YSI Pro2030) was used to collect information on temperature, dissolved oxygen (DO), and salinity, while a pH meter (SevenGo Mettler Toledo) was used to measure pH. Total suspended solids (TSS) was determined by collecting 500 ml of seawater from the sites, which were filtered through cellulose nitrate membrane filters (0.45 μm pore size, Whatman) that were then oven dried at 70 °C. The initial mass of the filter was subtracted from the mass after oven drying to obtain an estimate of the TSS among sites. Sedimentation rates were determined using sediment traps. The traps were deployed at the bommies at a depth of around 2 m. After 24 h, the contents of each trap were collected onto combusted, pre-weighed Whatman GF/F filters. Filters were dried at 60 °C to constant weight. Sedimentation rates were computed following the methods of *English, Wilkinson & Baker (1997)*. To determine water turbulence, 8 pre-weighed clod cards were placed at each bommie for 24 h and the percent difference in the dry weight of the clod cards before and after deployment was computed (*Doty, 1971*).

## Tetillid sponge characterization

The sponges were characterized in terms of external morphology. Spicule types were determined by bleach digestion followed by microscopic examination (*Hooper, 2003*). Tissue sections were prepared to examine the sponge skeleton structure. Diagnostic characters were matched to descriptions in the Thesaurus of Sponge Morphology (*Boury-Esnault & Rützler, 1997*), Systema Porifera (*Hooper & Van Soest, 2002*) and the work of *Santodomingo & Becking (2018)* to verify sponge identities.

Mitochondrial cytochrome oxidase 1 (CO1) gene sequencing was conducted to complement traditional morphological characters and to facilitate species identification. Genomic DNA was extracted using the PowerSoil DNA Extraction Kit (MO BIO) following the manufacturer's protocol. Amplification of the CO1 gene was done using the primers LCO1490 (*Folmer et al., 1994*) and COX1 R1 (*Rot et al., 2006*). The 25 μl PCR mix consisted of 1x PCR buffer (20 mM Tris-HCl pH 8.4, 50 mM KCl), 3 mM $MgCl_2$, 0.2 mM dNTPs, 0.4 μM each of forward and reverse primers, 1 unit Taq DNA polymerase (Invitrogen), and 30 ng of DNA. PCR amplification was conducted on a T100 Thermal Cycler (Bio-Rad, Munich, Germany) with an initial denaturation phase of 5 min at 94 °C, followed by 40 cycles of denaturation for 1 min at 94 °C, annealing for 1.5 min at 50 °C, elongation for 1.5 min at 72 °C, and a final elongation for 10 min at 72 °C (*Schuster et al., 2017*). Amplicons of ~1,500 bp or ~2,000 bp were gel-purified using the PureLink Quick Gel Extraction Kit (Invitrogen). Purified PCR products were sent to Macrogen Inc., South Korea, for direct sequencing. Sequences were aligned using ClustalW (*Thompson, Higgins & Gibson, 1994*) and trimmed using Gblocks (*Castresana, 2000*). Phylogenetic tree rendering using Bayesian inference was done using MrBayes v3.2.7a (*Ronquist et al., 2012*). Other CO1 sequences were obtained from *Szitenberg et al. (2013)*.

## Tissue sampling and DNA extraction

Six individuals each of *Cinachyrella* sp. and *Paratetilla* sp. were collected from the easternmost (bommie 22) and westernmost (bommie 15) bommies in December 2016 and April 2017. These bommies were selected because they were farthest away from each other. Sponge sampling was conducted with permission from the Philippines Department of Agriculture Bureau of Fisheries and Aquatic Resources under Gratuitous Permit No. 0125-17 and 0169-19. Sponges were sliced and fragments were washed with sterile seawater to remove any foreign macroscopic debris. To eliminate planktonic or loosely attached microorganisms and detritus, the cleaned fragments were rinsed with sterile calcium magnesium-free seawater (CMFSW) on a platform shaker at maximum speed for 10 min. After washing, fragments were further cut into ~0.5 g pieces and total DNA was extracted using PowerSoil DNA Extraction kit (MO BIO) following the manufacturer's protocol. Quality of extracted DNA was checked by agarose gel electrophoresis and concentration was determined using a Nanodrop spectrophotometer prior to 16S rRNA gene sequencing.

## Sequencing and microbial community analysis

Total genomic DNA extracted from 12 tetillid sponge samples (3 biological replicates per species per timepoint) were sent to Macrogen Inc., South Korea, for sequencing on the Illumina MiSeq platform. The V3–V4 region of the prokaryotic 16S rRNA gene was amplified using the primers Bakt_341F (5′-CCTACGGGNGGCWGCAG-3′) and Bakt_805R (5′-GACTACHVGGGTATCTAATCC-3′) (*Herlemann et al., 2011*). Raw sequence data were deposited in the NCBI Sequence Read Archive and can be accessed under BioProject accession number PRJNA596898. Demultiplexed paired end reads were analyzed using QIIME2 version 2018.11 (*Bolyen et al., 2019*; https://docs.qiime2.org). Raw data were imported and renamed according to QIIME2 sample data format Casava 1.8 paired-end demultiplexed fastq. Sequences were denoised by removing chimeric sequences and correcting amplicon errors using the DADA2 package (*Callahan et al., 2016*). Based on quality plots, reads were trimmed using the following parameters: -p-trim-left-f = 17; -p-trim-left-r = 21; -p-trunc-len-f = 290; and -p-trunc-len-r = 250. For taxonomic assignment, a naïve Bayes classifier was trained on SILVA version 132 (*Quast et al., 2012*; https://arb-silva.de) with reference sequences trimmed to the V3–V4 region. The trained classifier was applied to the representative sequences to assign taxonomy at 97% sequence identity. Sequence reads classified as chloroplast and mitochondria, as well as singletons, were removed using the commands "qiime taxa filter-table" and "qiime taxa filter-seqs." Amplicon sequence variant (ASV) counts were rarefied to the smallest sample size (20,818 sequences) prior to computation of alpha diversity metrics, such as Observed ASVs, Shannon, and Inverse Simpson. Alpha diversity metrics were computed using phyloseq (*McMurdie & Holmes, 2013*), Companion to Applied Regression (car) (*Fox & Weisberg, 2019*), and Ryan miscellaneous (Rmisc) (*Hope, 2013*). The Bray-Curtis community distance matrix was visualized using non-metric multidimensional scaling (NMDS) in vegan (*Oksanen et al., 2017*). Unrarefied ASV counts were used to calculate nonparametric Permutational Multivariate Analysis of Variance (PERMANOVA) using the Adonis method and Analysis of Similarity (ANOSIM) using 999 permutations for the

comparison of communities. Differentially abundant ASVs were identified using ANOVA-like differential expression (ALDEx2) analysis (*Fernandes et al., 2013*) with Welch's t test. All R packages were implemented in RStudio version 1.2.1335 (*RStudio Team, 2018*).

## Prediction of functional genes

Phylogenetic Investigation of Communities by Reconstruction of Unobserved States or PICRUSt2 (*Langille et al., 2013*) was used to predict functional gene abundance based on ASV taxon affiliations. The software was installed as a QIIME2 plugin. The commands "qiime fragment-insertion sepp" and "qiime picrust2 custom-tree-pipeline" setting the –p-max-nsti to 2 were used to generate functional prediction. The relative abundance profiles of predicted Kyoto Encyclopedia of Genes and Genomes (KEGG) ortholog (KO) genes were visualized using metaMDS. Linear discriminant analysis (LDA) of effect size or LEfSe was used to identify KOs that distinguish between the two species (*Segata et al., 2011*). KO terms with an absolute LDA >2.0 and alpha < 0.05 were considered discriminative features.

## Statistics

All data were tested for normality using Shapiro–Wilk test and homogeneity of variances using Levene's test. General Linear Models (GLM) implemented in Statistica v7 were used to examine the differences in mean density of tetillid sponges among bommies and across sampling periods, differences in mean density of the two species of tetillid sponge among bommies, and differences in environmental conditions among bommies. Results from GLM were further tested with Tukey's HSD post hoc test to see which bommies and sampling periods had significant differences. Kolmogorov–Smirnov tests were conducted to examine the differences in size frequency distributions of tetillid sponges across sampling periods per bommie. Statistical difference in alpha diversity between the microbial community of the two sponge groups was calculated using Welch's t test. A $p$-value <0.05 was considered significant. Data visualizations were produced using ggplot2 (*Wickham, 2016*), pheatmap (*Kolde, 2018*), and RColorBrewer (*Neuwirth, 2014*) in RStudio version 1.2.1335 (*RStudio Team, 2018*).

# RESULTS

## Distribution and size frequency of tetillid sponges on the reef bommies

Tetillid sponges were observed on all the bommies. The sponges were typically found covered in sediments and overgrown by turf algae, or in close interaction with other types of macroalgae, sponges, and corals (Fig. S2). The average density recorded over four years of monitoring ranged from $1.57 \pm 0.79$ to $4.46 \pm 3.60$ individuals per sq. m. per bommie. Sponge density was significantly greater on bommies 21 and 22 than on bommies 15, 16, and 19 (Fig. 1A; Tukey's HSD post hoc tests: $p < 0.05$). There was no change in sponge density over time (GLM: $F = 2.38$, $p = 0.09$) (Fig. 1A; Table S1). The average size of the tetillid sponges ranged from $2.75 \pm 2.11$ to $6.33 \pm 3.98$ cm, with very few sponges growing larger than 10 cm (Fig. 1B). A significant increase in sponge size frequency distribution was

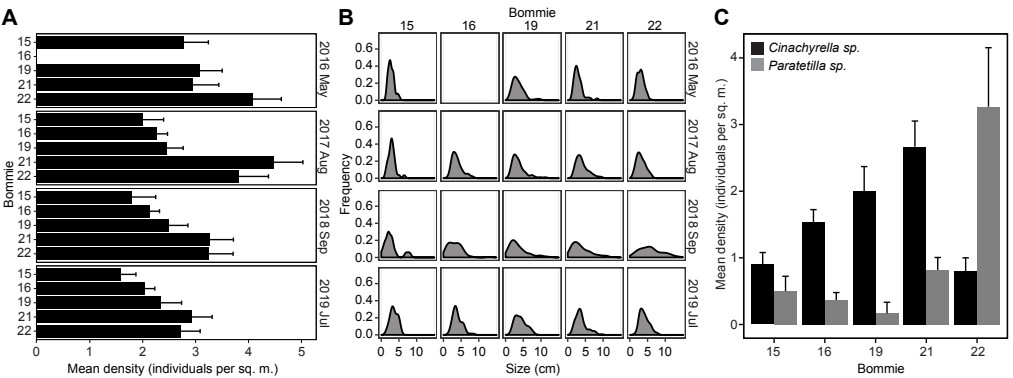

**Figure 1 Sponge population dynamics.** (A) Tetillid sponge population density on the coral bommies from 2016 to 2019. (B) Size frequency distribution of tetillids on the bommies. Bommie 16 was not included in the May 2016 survey. (C) Distribution of *Cinachyrella* sp. and *Paratetilla* sp. on the bommies based on a survey conducted in September 2019.

noticeable in September 2018 on all the bommies, except for bommie 15 (Fig. 1B; Table S2). Environmental parameters measured across the four bommies remained similar over the monitoring period (Table S3).

## Tetillid sponge morphology and sequencing

Tetillid sponges on the reef bommies were identified as *Cinachyrella* sp. and *Paratetilla* sp.. *Cinachyrella* sp. exhibited deeper hemi-spherical depressions called porocalices alternatively perforated by a number of small pores, or some oscular tubes, and had yellowish inner tissues (Fig. 2; Table S4). This sponge possessed spicules characterized as oxea, anatriaene, protriaene, sigmaspires and microoxea. *Paratetilla* sp. also had narrow hemi-spherical porocalices that were sometimes closed, and had brown internal tissues. This sponge possessed oxea, anatriaene, protriaene, sigmaspires, and microoxea spicules, as well as triradiate symmetrical rays. CO1 sequences from *Cinachyrella* sp. samples grouped with sequences from other *Cinachyrella* sp. while *Paratetilla* sp. samples clustered closely with sequences from *Paratetilla bacca* (Fig. S3).

Although the two species are difficult to distinguish based on their external morphology, a survey that examined internal tissue color of the sponges revealed that *Cinachyrella* sp. was distributed on all bommies at almost similar densities and was generally more abundant than *Paratetilla* sp., which was found at greater density only on the easternmost bommie (Fig. 1C).

## Diversity of tetillid sponge microbiomes

Sequencing of the 16S rRNA gene V3–V4 region on the Illumina Miseq platform returned a total of 2,068,178 reads. After sequence filtering, a total of 587,405 reads with an average of $48,950 \pm 14,495$ (mean $\pm$ standard deviation) reads per library were obtained from 12 libraries (6 *Cinachyrella* sp. and 6 *Paratetilla* sp. samples). 1,459 amplicon sequence variants (ASVs) were identified at 97% sequence similarity and classified into 35 phyla, 78 classes and 176 orders. Rarefaction curves reached a plateau at 20,818 sequences, indicating

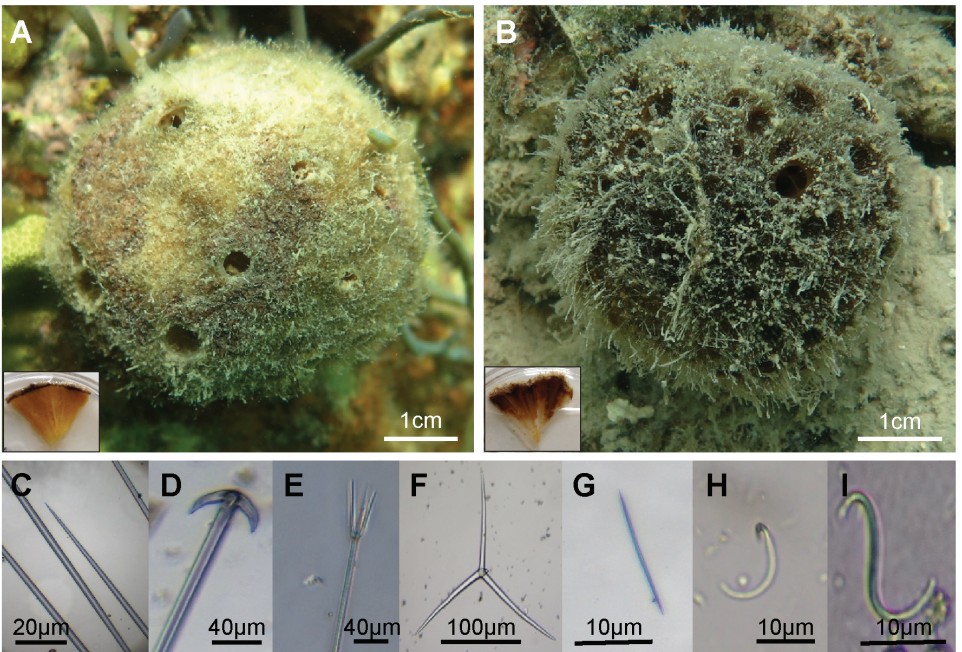

**Figure 2** **External morphology of *Cinachyrella* sp. (A) and *Paratetilla* sp. (B).** Hand-cut sections (insets) reveal the internal tissue color characteristic of each species. Spicules present in the tetillid sponges include megacleres oxea (C), anatriaene (D), protriaene (E), triradiate symmetrical rays (F), microscleres microxea (G), c-sigma (H), and s-sigma (I).

that the sequencing effort was sufficient to cover most ASVs in each sample (Fig. S4). No significant difference was observed in terms of the number of observed ASVs per species (Fig. 3A). However, *Paratetilla* sp. showed greater species richness and diversity compared to *Cinachyrella* sp., as indicated by significantly greater alpha diversity values based on the Shannon (Fig. 3B) and Inverse Simpson (Fig. 3C) indices.

The two sponge species possessed distinct prokaryotic microbial communities with only 11% (160 ASVs) of ASVs shared by both tetillids (Fig. S5). The difference in microbial community composition between the two species is apparent in the NMDS plot (Fig. 3D) and is statistically supported (PERMANOVA: $R^2 = 0.84118$, $p = 0.001$; ANOSIM: $R = 0.874$, $p$-value $= 0.001$) (Table S5). In contrast, no statistical difference was observed in the microbial communities of sponge individuals of the same species collected at different times (*Cinachyrella* sp., December 2016: April 2017: PERMANOVA: $R^2 = 0.35515$, $p = 0.1$; ANOSIM: $R = 0.4815$, $p$-value $= 0.1$; *Paratetilla* sp., December 2016: April 2017: PERMANOVA: $R^2 = 0.51471$, $p = 0.1$; ANOSIM: $R = 0.814$, $p$-value $= 0.1$) or from different bommies (*Cinachyrella* sp., Bommie 15: Bommie 22: PERMANOVA: $R^2 = 0.28559$, $p = 0.3$; ANOSIM: $R = 0.2593$, $p$-value $= 0.3$; *Paratetilla* sp., Bommie 15: Bommie 22: PERMANOVA: $R^2 = 0.287$, $p = 0.4$; ANOSIM: $R = 0.1852$, $p$-value $= 0.4$) (Table S5). This suggests that the microbiome associated with each sponge is species-specific.

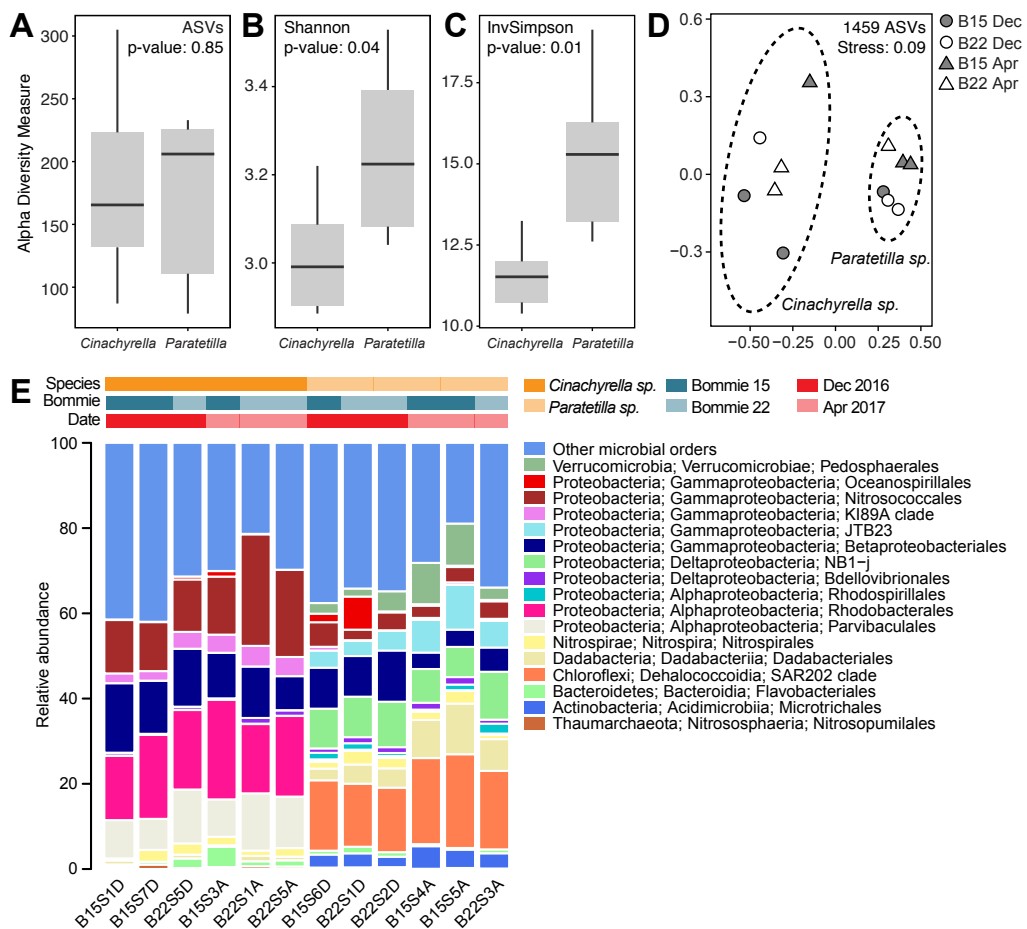

**Figure 3 Microbial community characteristics of the two tetillid sponges.** Comparison of alpha diversity indices including (A) observed ASVs, (B) Shannon and (C) Inverse Simpson between sponge microbial communities in the two tetillid sponge species. (D) Non-metric multidimensional scaling (NMDS) illustrating dissimilarity of microbial communities in *Cinachyrella* sp. and *Paratetilla* sp. individuals collected at different times (circle, Dec 2016; triangle, Apr 2017) and from different bommies (shaded, bommie 15; unshaded, bommie 22). The dashed ovals represent the 95% confidence area for each species. (E) Taxonomic assignments at order level showing the relative abundance of the 16S rRNA gene sequences of microbes associated with the two sponge species collected at different times from different bommies. Orders representing less than 0.4% of the total community are represented as "other microbial orders." Colored bars represent the relative abundance of microbial taxa in each replicate sample.

## Differentially abundant prokaryotes in *Cinachyrella* sp. and *Paratetilla* sp.

The *Cinachyrella* sp. microbiome was dominated by members of phylum Proteobacteria (90.28%), followed by Bacteroidetes (2.30%) and Nitrospirae (1.86%) (Fig. S6A). These included members of class Gammaproteobacteria (52.49%), Alphaproteobacteria (37.02%), Bacteroidia (2.29%) and Nitrospira (1.86%) (Fig. S6B). Amongst ASVs classifiable to the order level, the greatest proportion were affiliated with Rhodobacterales

(18.68%), Nitrosoccocales (16.09%), Betaproteobacteriales (12.19%), Parvibaculales (10.54%), KI89A clade (3.67%) and Nitrospirales (1.86%) (Fig. 3E).

On the other hand, the *Paratetilla* sp. microbiome was dominated by phylum Proteobacteria (59.52%), followed by Chloroflexi (17.95%), Dadabacteria (6.68%), Verrucomicrobia (5.32%), Actinobacteria (3.73%), Nitrospirae (2.20%), Patescibacteria (1.69%) and Bacteroidetes (1.39%) (Fig. S6A). This included members of class Gammaproteobacteria (29.53%), Alphaproteobacteria (18.26%), Dehalococcoidia (17.84%), Deltaproteobacteria (10.97%), Dadabacteriia (6.68%), Verrucomicrobiae (5.32%), Acidimicrobiia (3.68%), Nitrospira (2.20%), Parcubacteria (1.64%) and Bacteroidia (1.37%) (Fig. S6B). Amongst ASVs classifiable to the order level, the greatest proportion were affiliated with the SAR202 clade (17.84%), NB1-j (9.29%), Betaproteobacteriales (7.46%), Dadabacteriales (6.68%), JTB23 (6.11%), Pedosphareales (5.27%), Nitrosococcales (3.83%), Microtrichales (3.66%), and Nitrospirales (2.20%) (Fig. 3E).

Forty eight ASVs differed significantly in relative abundance between the two sponge species (Fig. 4) based on ALDEx2 analysis with Welch's test (*p*-value < 0.05). These differentially abundant ASVs showed no clear correlation with environmental variables at the collection site (Fig. S7). Twenty ASVs affiliated with class Gammaproteobacteria (10 ASVs), Alphaproteobacteria (6 ASVs), Dadabacteriia (1 ASV), Nitrospira (1 ASV), class Nitrososphaeria under Thaumarchaeota (1 ASV), and one unclassified bacterial ASV were found at relatively greater abundance in *Cinachyrella* sp. On the other hand, 28 ASVs belonging to class Gammaproteobacteria (8 ASVs), Alphaproteobacteria (7 ASVs), Nitrospira (3 ASVs), Deltaproteobacteria (2 ASVs), Acidimicrobiia (2 ASVs), Parcubacteria (2 ASVs), Dehalococcoidia (1 ASV), Dadabacteriia (1 ASV), one unclassified Proteobacteria ASV, and one unclassified bacterial ASV were found at higher relative abundance in *Paratetilla* sp.. Different ASVs of Nitrospiraceae, Betaproteobacterales EC94, Dadabacteriales, and Gammaproteobacteria KI89A clade were enriched in each sponge species.

## Predicted functional genes in tetillid-associated prokaryotes

Functional prediction was conducted using PICRUSt2, a software tool that predicts the functional profile of a microbial community based on 16S rRNA sequences (*Langille et al., 2013*). To improve accuracy of metagenome prediction, the weighted Nearest Sequenced Taxon Index (NSTI) value for the analysis was set to < 2.0 (*Langille et al., 2013*). NSTI reflects the relatedness of prokaryotic taxa in the sample to the closest available reference genome. Lower NSTI values (<2.0) indicate greater similarity to the reference, which results in a more precise prediction of functional genes (*Douglas et al., 2019*). However, it is important to note that PICRUSt2 is predictive and does not completely substitute for whole metagenome profiling (*Langille et al., 2013*; *Weigel & Erwin, 2017*). Nevertheless, it provides a starting point for understanding functions potentially represented within a microbial community.

PICRUSt2 predicted a total of 6892 KEGG ortholog (KO) genes from the microbial communities associated with the two sponge species. Of these, 6234 KOs (90.5%)

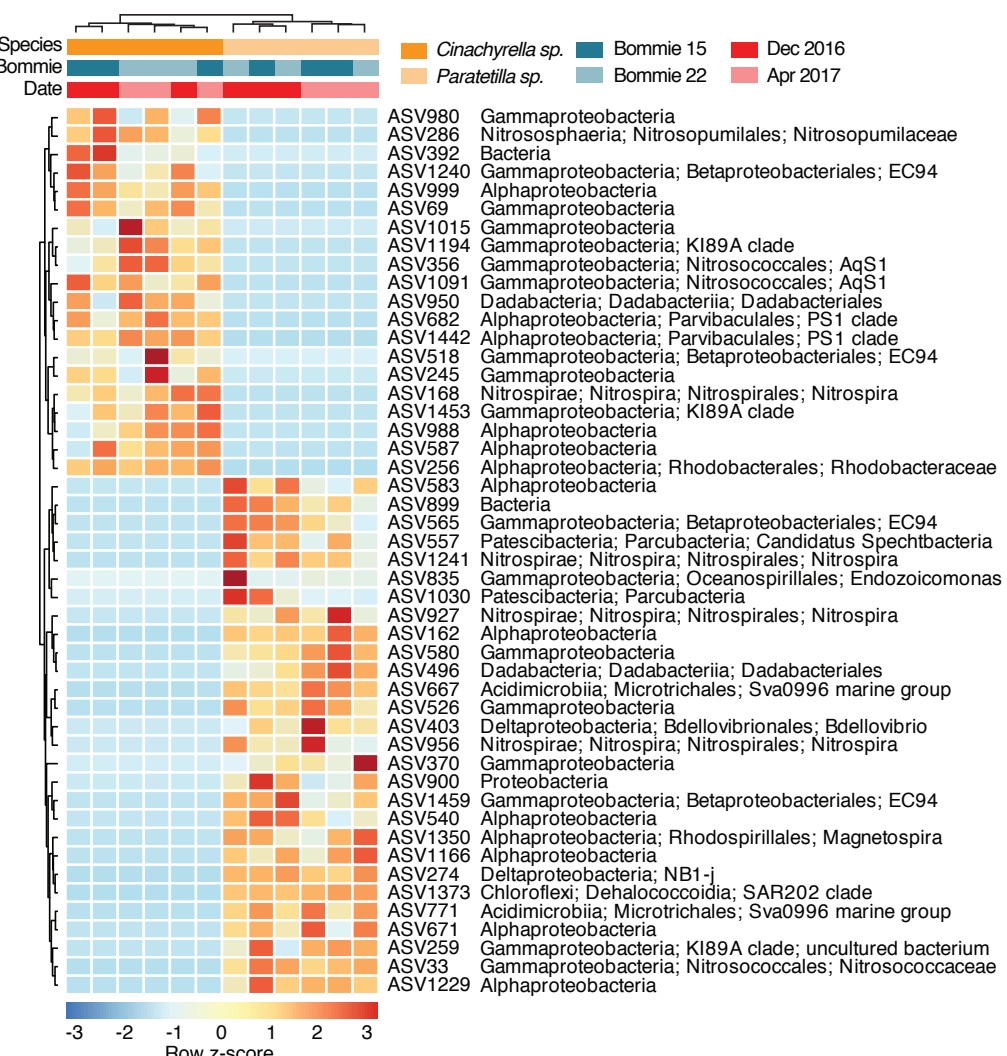

**Figure 4  Scaled heatmap of 48 differentially abundant microbial ASVs based on ALDEx2 analysis (*p*-value < 0.05).** Colors represent row z-scores of each microbial taxon (red, high; blue, low).

were present in both microbial communities, while 405 (5.9%) were present only in the *Cinachyrella* sp. microbiome and 253 (3.7%) were found only in the *Paratetilla* sp. microbiome. The predicted KO profiles of the microbial community of each sponge could be differentiated by NMDS (Fig. 5A) and these differences were statistically supported (*Cinachyrella* sp.: *Paratetilla* sp., PERMANOVA: $R^2 = 0.79417$, $p = 0.003$; ANOSIM: $R = 1$, $p$-value $= 0.002$) (Table S6). LEfSe analysis revealed an enrichment of KOs associated with ABC transporters, biosynthesis of secondary metabolites, fatty acid metabolism, glutathione metabolism, microbial metabolism in diverse environments, quorum sensing, sulfur metabolism, and terpenoid backbone biosynthesis in *Cinachyrella* sp.. KOs involved in bacterial chemotaxis, biosynthesis of amino acids and antibiotics, carbon fixation, citrate

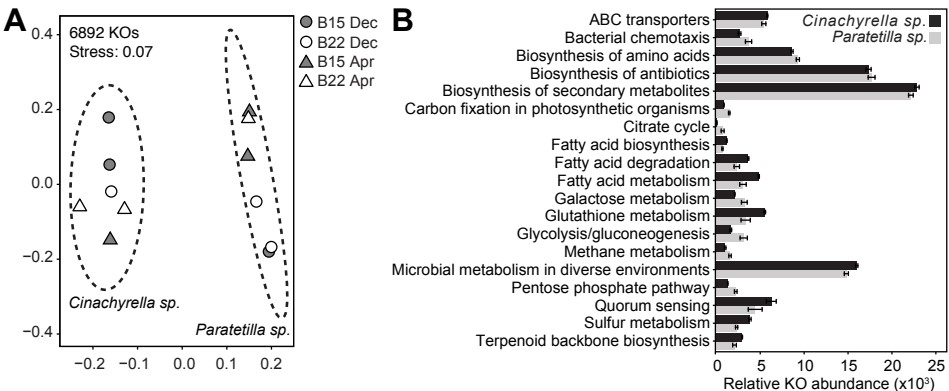

**Figure 5** **Functional gene predictions.** (A) Non-metric multidimensional scaling (NMDS) plot illustrating dissimilarity of the relative abundance profiles of PICRUSt2-predicted KEGG ortholog (KO) genes in the microbial communities associated with *Cinachyrella* sp. and *Paratetilla* sp. individuals collected at different times (circle, Dec 2016; triangle, Apr 2017) and from different bommies (shaded, bommie 15; unshaded, bommie 22). The dashed ovals represent the 95% confidence area for each species. (B) Average sums of the relative abundance of KOs in selected pathways. Only differentially enriched KOs in either species with LEfSe LDA > 2.0 and *p*-value < 0.05 were included.

cycle, galactose metabolism, glycolysis/gluconeogenesis, methane metabolism, and pentose phosphate pathway were enriched in *Paratetilla* sp. (Fig. 5B).

# DISCUSSION

## Tetillid sponge population dynamics

Tetillid sponges were abundant on the coral bommies and populations were stable over the course of 4 years of monitoring. The easternmost bommies (bommie 21 and 22) had the highest density of sponges compared to the others. These bommies face the Lingayen Gulf and are likely to be less exposed to strong wave action during typhoons. Most sponges exhibited an average diameter of about 3 to 6 cm, with very few growing to larger size. This further suggests that the sponges may be affected by various physical disturbances, such as grazing or predation, strong wave action, and sedimentation, all of which can limit growth or cause tissue loss or mortality. An increase in size frequency distribution was observed in 2018 on all bommies, except on bommie 15, although the cause remains unknown. The dynamics of the tetillid sponge population on these bommies are in contrast to that of *Xestospongia muta* in Florida, which showed an increase in abundance over the course of 6 years due to increased recruitment owing to suitable environmental conditions (*McMurray, Henkel & Pawlik, 2010*).

The stable population of the tetillid sponges suggests that individuals are long-lived and slow-growing. In fact, another tetillid sponge, *C. cavernosa,* has been found to increase in mean diameter by just 0.1–0.2 cm per year, with specific growth rates decreasing as sponge size increases (*Singh & Thakur, 2015*). Similarly, growth rate of settled buds of *Tethya citrina* decreased with sponge age (*Cardone, Gaino & Corriero, 2010*). Growth rates were affected by temperature, silicate concentration, dissolved oxygen, and the presence

of competitors (*Singh & Thakur, 2016*). On the bommies, tetillids were typically found interacting with or in close proximity to algae, corals, and other sponges. These organisms are known to produce allelochemicals and may inhibit sponge growth, similar to the reported growth-limiting effect of zoanthids on *C. cavernosa* (*Singh & Thakur, 2016*). Further studies to test the impact of other benthic reef organisms on the growth of tetillid sponges remain to be conducted.

## Species specificity of tetillid sponge prokaryotic microbial communities

The sponge-associated microbiota is host-specific (*Reveillaud et al., 2014*; *Thomas et al., 2016*). The prokaryotic microbial community of *Cinachyrella* sp. can be distinguished from that of *Paratetilla* sp., although these two sponges co-exist in the same biogeographic location and experience similar environmental conditions. The *Paratetilla* sp. microbial community composition reported here is similar to what has been reported for *Paratetilla* in other biogeographic regions, with the dominance of Proteobacteria (Alpha- and Gammaproteobacteria), Chloroflexi, and Actinobacterial taxa (*Thomas et al., 2016*; *De Voogd et al., 2018*; *Cleary et al., 2019*). The *Cinachyrella* sp. microbiome from this study also showed some common taxa with other *Cinachyrella* microbiomes, which are dominated by Proteobacteria, Bacteroidetes, Cyanobacteria, and Actinobacteria (*Cleary et al., 2013*; *Cuvelier et al., 2014*; *Cleary, Polónia & De Voogd, 2018*). In addition, differentially abundant bacterial taxa found in both tetillid sponges have previously been reported as symbionts of other sponge species. For example, Nitrospiraceae was dominant in *Rhabdastrella globostellata* (*Steinert et al., 2016*), Betaproteobacteria EC94 was abundant in *Callyspongia* sp. (*Steinert et al., 2016*), and Gammaproteobacteria K189A was abundant in *Petrosia ficiformis* (*Burgsdorf et al., 2014*).

The microbial community in each sponge species remained similar in samples taken during different times and from different bommies. Moreover, no correlation was found between differentially abundant prokaryotic taxa and measured environmental variables. This mirrors findings from other studies in marine sponges that suggest that microbial communities are shaped by host identity (*Chambers et al., 2013*; *Naim et al., 2014*; *Souza et al., 2016*; *Steinert et al., 2016*). The sponge microbiome has been shown to be stable across individuals taken from different sampling locations or depths (*Pita et al., 2013*; *Reveillaud et al., 2014*) and can even withstand moderate pollution stress (*Gantt, López-Legentil & Erwin, 2017*; *Baquiran & Conaco, 2018*).

Bacteria–bacteria interactions may play a role in structuring the sponge microbial community. Some sponge-associated bacteria can inhibit the growth of other members of the community through the production of various compounds and regulatory signals (*Esteves, Cullen & Thomas, 2017*; *Gutiérrez-Barranquero et al., 2017*). For instance, *Bdellovibrio,* which is enriched in *Paratetilla* sp., is an active predator of other microorganisms and produces compounds that attack the cell walls of other bacteria (*Beck et al., 2004*). Other sponge-associated prokaryotes are attracted to sponge host-derived compounds, indicating an active role of the microbes in initiating species-specific partnerships (*Tout et al., 2017*; *Lurgi et al., 2019*). This is supported by the predicted

abundance of genes related to bacterial chemotaxis in both tetillid sponge species. On the other hand, bacteria with reduced genomes may exist in the community as ectosymbionts or parasites, relying on the biosynthetic capabilities of the host-associated microbiome (*Nelson & Stegen, 2015*). An example of this is Parcubacteria, which was detected in *Paratetilla* sp.

## Predicted functions of tetillid-associated prokaryotes

Sponge-associated prokaryotes fulfill functions that provide important benefits to the host and can also influence ecosystem health and function (*Taylor et al., 2007*; *Bell, 2008*; *Thomas et al., 2010*). In the present study, genes critical for metabolism, defense, and stress response were predicted to be present in the microbiomes associated with *Cinachyrella* sp. and *Paratetilla* sp.. Various differentially abundant ASVs in the tetillid sponges were affiliated with taxa known to be involved in the nitrogen cycle. Nitrifying prokaryotes that transform ammonia to nitrite, such as Nitrosopumilaceae (*Li et al., 2014*; *Feng et al., 2016*), the AqS1 group of Nitrosococcaceae (*Rua et al., 2015*; *Feng & Li, 2019*), and the SAR202 clade of phylum Chloroflexi (*Morris et al., 2004*; *Mincer et al., 2007*), were detected in the tetillids. Members of *Nitrospira,* some of which are enriched in *Paratetilla* sp., may contribute to the conversion of nitrite into nitrate (*Hentschel et al., 2002*; *Daims & Wagner, 2018*). Members of Proteobacteria and Bacteroidetes, which were also identified in the tetillids, may play a role in denitrification, which removes excess nitrate from the sponge tissues (*Hoffman et al., 2009*; *Feng & Li, 2019*). The potential co-existence of nitrification and denitrification functions in the tetillid sponge microbiota suggests that affiliated prokaryotes can adapt to shifts from aerobic to anaerobic conditions inside the sponge (*Schlappy et al., 2010*). In addition to supplying the nitrogen requirements of the holobiont, nitrogen metabolism by sponge-associated microbes may also benefit other biota, such as macroalgae and other organisms, in the surrounding area (*Davy et al., 2002*).

Genes related to sulfur metabolism, including sulfur oxidation and sulfate reduction, were predicted to be present in the tetillid microbiomes. These two processes may be coupled, as has been demonstrated in the cold water sponge *Geodia barretti* (*Jensen et al., 2017*). Sulfur oxidation is a potential mechanism for the removal of toxic metabolic end-products, such as hydrogen sulfide, that are produced by the sponge host. The existence of an anoxic micro-ecosystem in the tetillid sponges is further supported by the presence of sulfate-reducing bacteria (SRB), such as members of Deltaproteobacteria and Dadabacteria (*Wasmund, Mußmann & Loy, 2017*; *Hug et al., 2015*).

Genes in key biosynthetic pathways were predicted to be present in both tetillid sponge microbiomes. The microbiome of *Paratetilla* sp., in particular, was enriched for genes in the carbon fixation pathway, pentose phosphate pathway, galactose metabolism, glycolysis, and citrate cycle. Translocation of fixed carbon to the sponge would provide a valuable source of alternative nutrition for the host, analogous to photosynthates from autotrophic microbes (e.g., Cyanobacteria) (*Kandler et al., 2018*).

The *Cinachyrella* sp. microbiome was enriched for fatty acid metabolism genes. This suggests that this species may produce a diverse array of fatty acids (*Rod'kina, 2005*), which could serve as a potential energy store or as building blocks for bioactive compounds. The enrichment of the terpenoid biosynthesis pathway in the *Cinachyrella* sp. microbiome

further suggests an active involvement in secondary metabolite production, as has been reported for other species of sponges (*Cleary, Polónia & De Voogd, 2018*; *Steinert et al., 2019*). Secretion of secondary metabolites, including terpenoids, may have allelopathic effects on other organisms and may contribute to the differential distribution of the two sponge species. However, a detailed assessment of the secondary metabolites produced by each tetillid sponge remains to be conducted.

Genes related to antibiotic production and glutathione synthesis were also predicted to be present in the *Cinachyrella* sp. and *Paratetilla* sp. microbiomes. *Endozoicomonas*, which is abundant in *Paratetilla* sp., can produce quorum sensing metabolites and demonstrates antimicrobial properties against potentially harmful microbes (*Esteves, Cullen & Thomas, 2017*; *Mohamed et al., 2008*; *Morrow et al., 2015*; *Rua et al., 2014*). We hypothesize that the abundance of protective genes in the tetillid sponge-associated symbionts may be an adaptation to stressful conditions, such as high temperatures, high sedimentation rates, and eutrophic waters, that are frequently encountered in the reef lagoon.

Although the microbial community composition of the two tetillids were distinct from each other, it is interesting to note that the predicted functions represented in the microbiomes were similar. This observation supports the concept of functional equivalence, wherein the microbiomes of phylogenetically divergent hosts that occupy similar functional niches may convergently evolve to perform similar core functions, likely through the process of horizontal gene transfer (*Fan et al., 2012*). Functional convergence of the microbial communities in tetillid sponges indicates the presence of core functions that may be critical for their health and survival on the reef, and may partly explain the stability of the sponge populations on the bommies. Differentially enriched functions, on the other hand, may indicate species-specific adaptations influenced by host metabolism or chemistry (*Cleary et al., 2015*; *Steinert et al., 2019*).

## CONCLUSION

In this study we identified two tetillid species, *Cinachyrella* sp. and *Paratetilla* sp., that are found in abundance on coral bommies within a reef lagoon. The density and size frequency of the sponge populations remained relatively stable over the course of the monitoring period of approximately four years, although *Cinachyrella* sp. was dominant on more bommies. The sponges host distinct microbial communities, supporting the idea of species-specificity of the sponge microbiome. However, predicted functions represented within the microbiota of the two species present a large overlap, indicating functional equivalence of the communities driven by prevailing environmental conditions at the site. Nevertheless, certain functions could be distinguished as differentially enriched between species, particularly pathways related to carbon, sulfur, fatty acid, and amino acid metabolism, cellular defense, and stress response. These likely indicate microbiome-specific adaptations to host metabolism and may influence the interactions of the sponges with other biota on the bommies. Further validation of the functional profiles of the tetillid sponge-associated microbiota using metagenome or metatranscriptome approaches are warranted in order to verify the genes that are present and expressed, as well as the microbial players contributing to functions of interest.

## ACKNOWLEDGEMENTS

The authors thank Fernando Castrence Jr., Renato Adolfo, Ronald De Guzman, Ben Jack Gabuay, Robert Valenzuela, Francis Kenith Adolfo, and other staff of the Bolinao Marine Laboratory for their invaluable assistance in the field surveys.

### Funding

This study was funded by the University of the Philippines System Enhanced Creative Work and Research Grant (ECWRG 2018-1-002) to Cecilia Conaco and by the Department of Science and Technology Philippine Council for Agriculture, Aquatic and Natural Resources Research and Development (QMSR- MRRD-MEC-295-1449) to Patrick C. Cabaitan and Cecilia Conaco. There was no additional external funding received for this study. The funders had no role in study design, data collection and analysis, decision to publish, or preparation of the manuscript.

### Grant Disclosures

The following grant information was disclosed by the authors:
University of the Philippines System Enhanced Creative Work and Research Grant: ECWRG 2018-1-002.
Department of Science and Technology Philippine Council for Agriculture, Aquatic and Natural Resources Research and Development:  QMSR- MRRD-MEC-295-1449.

### Competing Interests

The authors declare there are no competing interests.

### Author Contributions

- Jake Ivan P. Baquiran performed the experiments, analyzed the data, prepared figures and/or tables, authored or reviewed drafts of the paper, and approved the final draft.
- Michael Angelou L. Nada performed the experiments, analyzed the data, authored or reviewed drafts of the paper, and approved the final draft.
- Niño Posadas and Dana P. Manogan analyzed the data, authored or reviewed drafts of the paper, and approved the final draft.
- Patrick C. Cabaitan and Cecilia Conaco conceived and designed the experiments, analyzed the data, prepared figures and/or tables, authored or reviewed drafts of the paper, and approved the final draft.

### Field Study Permissions

The following information was supplied relating to field study approvals (i.e., approving body and any reference numbers):

Sponges were collected by SCUBA diving in Bolinao, Pangasinan, Philippines, with permission from the Philippines Department of Agriculture Bureau of Fisheries and Aquatic Resources (DA-BFAR Gratuitous Permit No. GP-0125-17 and GP-0169-19).

## DNA Deposition

The following information was supplied regarding the deposition of DNA sequences:

Raw sequence data are available at the NCBI Sequence Read Archive, BioProject: PRJNA596898.

## Data Availability

Raw data are available at FigShare: Baquiran, Jake Ivan; Conaco, Cecilia; Nada, Michael Angelou; Posadas, Niño (2020): Supplementary data for *Cinachyrella* sp. and *Paratetilla* sp. population structure and microbiome study. figshare. Dataset. https://doi.org/10.6084/m9.figshare.11664066.v3.

## Supplemental Information

Supplemental information for this article can be found online at http://dx.doi.org/10.7717/peerj.9017#supplemental-information.

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
