# Peer review of "Population structure and microbial community diversity of two common tetillid sponges in a tropical reef lagoon"

_PeerJ, doi:10.7717/peerj.9017_

## Round 0.1 · original submission · Minor Revisions

I now have three reviews from experts in sponge-microbial associations, and I am pleased to report that they are positive and all three recommend minor revisions. Congratulations!

Please note that all three reviewers indicate specific edits that the authors should undertake. Please provide a separate document that lists the reviewer comments and author responses in a point-by-point manner.

Reviewer 1 ·

Basic reporting

In this study Baquiran and colleagues examine the population dynamics and microbial community composition of moon sponges on coral bommies in the Philippines. They found the population structure to be stable over the course of the 4-yr study, with the two species of moon sponge harbouring distinct microbiomes. However, there was an overlap in their functional profiles as determined by PICRUSt2. On initial passthrough of the abstract there was some expectation that there would be metagenomic data to support the convergence of functions identified between the two species. While the authors do a good job of identifying the limitations in the methods section, it should be made clear from the start that conclusions are drawn from PICRUSt2. In addition, the first line of the abstract states that, “Sponges are predicted to dominate future reef ecosystems influenced by anthropogenic stressors and global climate change” however, the introduction is lacking any information to directly support this statement. There is a growing body of literature that can be used see Bell et al. 2018 DOI: 10.1093/biosci/biy142 as a starting place. Other suggestions pertaining to the introduction can be found in the general comments.

Figures and tables
• Suggest Fig 1 becomes Supplementary and Figure 4 be split. Perhaps (D) on its own then the fonts can be enlarged for all graphics, which are currently hard to read as is.
• Supp Fig 2 – very hard to read tree, increase font size. In addition, suggest red font is replaced with another color.
• Supp Table 5 and 7-11 are not mentioned within text. In addition, all tables and figures need to be in cited in sequential order. For example, Supp table 6 is cited before Supp table 1, please correct throughout.

Raw data
• Raw sequencing data should be submitted to the NCBI SRA or equivalent prior to acceptance.

Experimental design

The research question was well defined and the methods logical, though I have a few comments/suggestions:
• What was the justification for where the sponge samples for micro were collected (i.e. why bommie 22 & 15)? Also, it is not clear from the NMDS which samples came from 22 vs 15.
• You have used DADA2, but are reporting OTUs classified at 97% similarity and not ASVs, is there a reason why?
• While there was no significant difference in environmental variables between the sampling times, the dataset collected provides the unique opportunity to examine if any of the environmental variables measured are correlated with microbiome structure in these sponges. This can be done using the envit test in vegan or you could use DISTLM/dbRDA and would provide an extra element to the study.

Validity of the findings

This manuscript contributes to the growing body of literature on the microbiomes associated with sponges, as well as information on sponge population dynamics over multiple years, which is a strength of the study. Authors have applied statically sound methods; the data is robust and the paper itself well written.

Additional comments

Line 58: Instead of just citing a review, this statement should also include a reference that examines the microbiomes of sponges. Work from the global sponge microbiome project would be a good option, Thomas et al. 2016.
Line 67: specifically, what environmental conditions? Suggest expanding on this topic.
Line 262: ‘individuals’ is not needed
Line 306: insert ‘,’ after (16.09%)
Line 337: it might be worth clarifying what the NSTI score relates to and why the chosen cut-off is acceptable.
Line 386: while the statement is factually correct, the connection between it and your study is lacking given no seawater or sediment was examined.
Line 392: given the substantial number of sponge-microbiome studies available, there should be more than one that backs up your findings of these dominant taxa.
Line 412: what is the relevance to this study? Have SRCR been identified in moon sponges? If no direct link can be formed, this paragraph should be removed.
Line 456: remove ‘,’ after ‘addition’

·

Basic reporting

The manuscript by Baquiran et al is a nice piece of work on the population structure and of two related sponge species and their associated prokaryotes. It is based on a very complete dataset (although you might advocate that seawater samples should also have been included as a reference for the prokaryotic communities) and analyses are well done. The story is well written, was easy to follow and well... it is just a nice piece of work.
Detailed comments are listed per line number in the last box “general comments for the authors”

Experimental design

Experimental design is well done except perhaps the missing seawater samples and some explanation is needed what is the exact target of the 16S rRNA gene primers used and why not both COI and 28S are sequenced from both sponges.

Validity of the findings

The findings are valid. Speculation is well indicated and underpinned. I do find the section on the predicted gene functions rather long as it remains a prediction and the predictions described do not add very much to our insights. Now they overtake a little bit your own data.

Additional comments

l33+34: not clear how to interpret the numbers or what they exactly refer to
l35: what is the "moon sponge"? It is for the reader at this point not clear that they refer to both species. Or even more?

l78: it should be made clear that "moon sponges" is the colloquial name??? Now it is mixed with classical taxonomic terms. It is ok to use the colloquial terms, but it needs to be specified which taxonomic groups are covered by the colloquial name

l90: From the aim of the study it is not clear how this discriminates the study from the studies of Cuveliers and Chambers.

l160: was template DNA diluted to a certain concentration or was the DNA extraction solution used undiluted?

l184: No water samples were included as reference for the bacterial communities. It would have been better if they had, but I think it is not strictly necessary as it has been proven by many scientists that sponge-associated prokaryotic communities are different from the seawater and that seawater communities do not really interfere with the conclusions taken.

l188: The 16S rRNA gene primer names suggest that only bacterial genes are amplified. Yet throughout the manuscript "microbes" "microorganisms" etc. are used. Microbes comprise the 3 domains of life, so better to change throughout the manuscript to "bacteria".
In the results I see that also Thaumarchaeota were recovered. If the primers target both bacteria and archaea, better use the term "prokaryotes" to refer to both.

l244: Would it be possible to indicate the moon sponge species for Fig 2A-F?
Actually.... although Fig.2 is prettier than suppl. fig 1, I think suppl. fig. 1 is a better figure for the paper than Fig.2

Why are the Cinachyrella and Paratetilla specimens not in both trees in suppl. Fig. 2? Now we can see how they relate to other sponges, but not how they are related to each other rather than only on the morphological characteristics described.

l289: what do the dashed ovals in Fig. 4b signify?

l366: "a change in size frequency..." better include the direction of change. Now it remains a bit mysterious

l434 and on: I don't have fundamental objections against these functional predictions, but think we don't learn so much from this section (85 lines), while the rest of the manuscript is really well covered by solid data.
Also 50% of the conclusion is based on these functional predictions. To my opinion your other results are more interesting

Reviewer 3 ·

Basic reporting

Please introduce the common name “moon sponges” both in the abstract (line 35) and the introduction (line 84). Presumably the name refers to both (all?) tetillid species, but this is not explicitly stated.

In fact, there is some general confusion about two species that only becomes clear (to me at least) later on in the manuscript. Namely, that the two species were not distinguished in most of the field monitoring due to there similar superficial morphology. It would be worth to clarify this earlier in the text.

Can sponge species name be added to Fig. 2 – or is not confirmed for these photos?

Line 408: Typo – change “Lopex” to “Lopez”

Lines 410-418: Interest work is summarized here, but a lack of sponge genome data in the present make this paragraph seem a bit tangential (i.e. the data herein do not inform this hypothesis).

Experimental design

Clear research questions and solid study design: multiple years of monitoring, replication in microbiome characterization, and careful taxonomic and phylogenetic analysis of hosts.

Validity of the findings

The data is robust and statistically sound.

Additional comments

The authors present a solid, well-written manuscript that adds to fields of sponge microbiology and coral reef biology. All aspects are well done: from field surveys, to microbiome characterization and comparison, to host taxonomic and phylogenetic analyses. Accordingly, the manuscript yield important and well-supporting new findings. Aside from a few small comments above, the manuscript is in great shape and worthy of publication. I congratulate the authors on a thorough study.

---

## Round 0.2 · accepted · Accept

Thanks for responding to the reviewers' comments and revising your manuscript accordingly.